# Tetanus Toxin Fragment C: Structure, Drug Discovery Research and Production

**DOI:** 10.3390/ph15060756

**Published:** 2022-06-17

**Authors:** Caroline Bayart, Angélique Mularoni, Nada Hemmani, Soumeya Kerachni, Joachim Jose, Patrice Gouet, Joseph Paladino, Marc Le Borgne

**Affiliations:** 1Analytical Sciences, Sanofi, 31/33 Quai Armand Barbès, 69250 Neuville-sur-Saône, France; caroline.bayart@outlook.com (C.B.); joseph.paladino@sanofi.com (J.P.); 2Small Molecules for Biological Targets Team, Centre de Recherche en Cancérologie de Lyon, Centre Léon Bérard, CNRS 5286, INSERM 1052, Université Claude Bernard Lyon 1, Univ Lyon, 69373 Lyon, France; nada.hemmani@etu.univ-lyon1.fr (N.H.); soumeya.kerachni@etu.univ-lyon1.fr (S.K.); 3Institute of Pharmaceutical and Medicinal Chemistry, PharmaCampus, Westfälische Wilhelms-Universtität Münster, 48149 Münster, Germany; joachim.jose@uni-muenster.de; 4UMR 5086, Molecular Microbiology and Structural Biochemistry, CNRS/Université Lyon 1, 69367 Lyon, France; patrice.gouet@ibcp.fr

**Keywords:** tetanus toxin fragment C, structure, production, uses, vaccine, neuronal protection, CNS delivery, immunogenicity, carrier protein, fusion protein

## Abstract

Tetanus toxoid (TTd) plays an important role in the pharmaceutical world, especially in vaccines. The toxoid is obtained after formaldehyde treatment of the tetanus toxin. In parallel, current emphasis in the drug discovery field is put on producing well-defined and safer drugs, explaining the interest in finding new alternative proteins. The tetanus toxin fragment C (TTFC) has been extensively studied both as a neuroprotective agent for central nervous system disorders owing to its neuronal properties and as a carrier protein in vaccines. Indeed, it is derived from a part of the tetanus toxin and, as such, retains its immunogenic properties without being toxic. Moreover, this fragment has been well characterized, and its entire structure is known. Here, we propose a systematic review of TTFC by providing information about its structural features, its properties and its methods of production. We also describe the large uses of TTFC in the field of drug discovery. TTFC can therefore be considered as an attractive alternative to TTd and remarkably offers a wide range of uses, including as a carrier, delivery vector, conjugate, booster, inducer, and neuroprotector.

## 1. Introduction

Tetanus caused by the tetanus toxin (TT) is a fatal illness, which despite the existence of a vaccine, led to an estimated 34,684 deaths in 2019 [1]. TT is a neurotoxin produced by *Clostridium tetani*, a Gram-positive pathogenic bacterium, mainly found in soil and the gastrointestinal tracts of animals. TT induces the inhibition of neurotransmitter release, leading to spastic paralysis in a four-step process [2]. First, TT binds to specific receptors, mainly composed of lipids and gangliosides, found at the neuromuscular junction (NMJ) [3]. Another receptor is reached by TT after these first bindings: a protein receptor responsible for its internalization (second step). This double receptor binding is responsible for the high affinity between TT and nerve cells. Third, TT is then transported into the cell body via axonal retrograde transport. In the last step, the proteolytic cleavage by TT of the VAMP/synaptobrevin, a neuronal substrate, leads to the inhibition of neurotransmitter release [4]. All of these biological properties can be distinct associated parts of the TT structure. TT is a 150.7 kDa protein composed of a 52.4 kDa light chain and a 98.3 kDa heavy chain linked by a disulfide bond [5,6].

The light chain, also named fragment A, has a zinc-dependent proteolytic activity responsible for synaptobrevin cleavage [3]. The heavy chain is composed of the fragments B and C (Figure 1). The 46.7 kDa fragment B is located at the N-terminus of the heavy chain and is responsible for toxin cell penetration. Finally, TT protein is completed by the 51.6 kDa fragment C, located at the C-terminus of its heavy chain. This latter is involved in neuronal cell binding.

The three fragments were discovered in 1974 during a study in which TT was enzymatically digested with papain, yielding fragments B and C [7]. Based on this study, several authors decided to test the properties of each fragment in order to identify their role in the TT protein. Tetanus toxin fragment C (TTFC) was rapidly in the limelight as it appeared to be non-toxic and presented interesting immunological features and neuronal binding properties. The usual terms and abbreviations used for TTFC are listed in Table 1.

In view of the numerous potential therapeutic applications of TTFC, especially in the fields of vaccination and neurology, different methods of production, characterization and evaluation were developed. This review, therefore, offers an overview of TTFC by describing its structure and its numerous properties regarding immunology and neuroprotection but also reflecting the different methods used for producing it. Several examples of therapeutic applications of TTFC and closely related products (e.g., conjugates, fusion proteins) are also reported.

## 2. Structure and Association with Receptors

The native structure of TTFC is available at the Protein Data Bank under the identifier 1AF9 [8]. It is composed of the last 452 amino acid residues of TT from K865 to D1315 and contains two adjacent domains, which display a lectin-like jelly roll motif and a β-trefoil motif, respectively (Figure 2). Since the publication of this first structure in 1997, other structures obtained by X-ray diffraction have also been published (1A8D, 1D0H, 1DFQ, 1DIW, 1DLL, 1FV3, 1FV2, 7CE2) [9].

The amino-terminal jelly roll domain is composed of two curved seven-stranded β-sheets wrapped together to form a convex solvent-exposed face (strands β2, 1, 16, 7, 12, 13, 14) and a concave one (strands β3, 6, 15, 8, 9, 10, 11) (Figure 2A). The carboxyl-terminal β-trefoil domain is composed of strands β17, 21, 22, 26, 27, 30 for the β barrel and strands β18, 19a, 23, 25, 28, 29 for the β hairpin triplet (Figure 2B). This second motif can be observed in proteins involved in recognition or binding mechanisms such as histactophilin, interleukin 1α and 1β or fibroblast growth factors [8]. Site-directed mutagenesis as well as crystallization experiments made by soaking TTFC with various carbohydrates have shown that the β-trefoil domain plays a predominant role in the attachment to ganglioside receptors [2,10]. These complex molecules are sialic acid-containing glycosphingolipids highly present at the outer surface of neuronal cell membranes. The accessible moiety of ganglioside is made of an oligosaccharide of 2 to 4 hexoses to which one or more sialic acids are attached. Two distinct ganglioside binding sites have been identified at the surface of the β-trefoil domain, i.e., (i) a lactose binding site named W-pocket where the oligosaccharide core structure can bind [2,10,11,12,13,14] and (ii) a sialic acid binding site named R-pocket, which can accommodate a di- or a trisialic acid [15]. The two pockets are approximately 15 Å apart.

The R-pocket is delineated by segments 1143–1148, 1213–1216, 1226–1235 and 1271–1282. Asp1147 and Arg1226 are considered to be key binding residues [12]. The W-pocket contains binding residues Asp1222, His1271, Trp1289, Tyr1290, His1293, Phe1218 and Thr1270 [2,14,16], Trp1289 being essential for maintaining the shape of the pocket [12].

TTFC binds to gangliosides from the G1b series, which are predominant in the brain [12,17]. The tetrasaccharide core of these gangliosides is composed of Gal(β1–3)GalNac(β1–4)(NeuAc(α2–3))Gal(β1–4)Glc(β1), with Glc(β1) linked to membrane-embedded ceramide, to which at least two sialic acids are attached. A disialic acid linked to the internal galactose seems to be important for the association. Surface plasmon resonance experiments have revealed that TTFC strongly binds to disialylganglioside G_D_1b and slightly less to trisialylganglioside G_T_1b [17,18]. Affinity was not significant for G_Q_1b, which has a diasilic acid attached to each galactose (Figure 3). Further investigations using glycan array and solid-phase binding analysis suggest that TTFC can have a dual attachment mode and can concomitantly bind to one ganglioside at the R-pocket and one at the W-pocket [14].

In addition, it is hypothesized that the binding of TTFC to neurons could involve both gangliosides and a protein receptor. A 15–20 kDa putative receptor (p15) has been identified and different molecules have been tested for their affinity to bind TTFC [19]. In particular, a peptide Tyr-Glu-Trp and doxorubicin have shown strong interactions with TTFC [12,13]. The tripeptide binds to the R-pocket in a competitive manner with disialyllactose, suggesting that this site can interact with both proteins and carbohydrates.

In conclusion, structural data are key to understanding TTFC binding to neurons and its internalization into the central nervous system (CNS).

## 3. TTFC Properties and Uses: CNS Delivery and Immunogenicity

### 3.1. TTFC Neurological Properties

#### 3.1.1. Permeability and CNS Delivery

TTFC was reported to bind to motor neurons (MN) [19]. TTFC contains distinct binding sites, which are recognized by neuronal receptors. This is the first step of its transport into the CNS. It especially binds to polysialylgangliosides G_D_1b and G_T_1b (Figure 3), which are present in the CNS. Their role is to regulate protein activities at the lipid membrane [20]. After ganglioside recognition, TTFC is internalized by MN at the NMJ [21]. As it has the ability to form pores in lipid vesicles, the third step of its transfer is its translocation through the lipid membrane. This translocation depends on the ganglioside content but also on the pH of the membrane [11,22]. It finally reaches the neuronal perikaryon in the CNS, where it can target the recognition and the catalytic cleavage of neuronal substrates [11]. Owing to its ability to reach the CNS and to specifically affect neurons, TTFC seems to be a promising candidate as a vector for neuronal disease treatment [22].

Indeed, used as a fusion protein or carrier after chemical linkage, TTFC may be useful to deliver either proteins or drugs into the CNS to cure neurodegenerative diseases. After intramuscular injection or systemic administration, such chemically altered TTFC were shown to enhance protein delivery in neurons by 1000-fold in a mouse model [23]. TTFC was also reported to improve both the neuronal uptake and the distribution of brain-directed therapeutics [23,24]. As the blood–brain barrier (BBB) is relatively difficult to cross, drugs are directly injected into the brain or into cerebrospinal fluid. Using TTFC as a vector for protein delivery could simplify the route of injection. Indeed, a fusion protein administrated by intramuscular injection was detected on endosomal and synaptic vesicles [24]. These two locations are interesting as the synaptic membrane can be targeted for neurotransmitter-related protein or trophic factor delivery, and endosomal and lysosomal locations can be targeted for metabolic enzyme delivery [23]. This method has been used successfully to deliver proteins to MN [24]. Importantly, several studies have shown that a TTFC/cytokine fusion had no effect on neuronal properties or cytokine protection properties [25].

Fusion proteins with TTFC can also be used to study synaptic connections. Fusion with a labeled protein (e.g., GFP, green fluorescence protein) allows transportation of the protein through synapses in a non-toxic way. Such a tool can be used to study the mechanisms of TTFC internalization and to map the neuronal network [26]. Indeed, TTFC is able to fuse with other molecules without altering its trans-synaptic and retrograde properties. This type of fusion has led to the successful study of intra- and inter-neuronal trafficking in vivo [27].

The unique properties of TTFC have also been exploited to construct a TT-derived peptide (Tet1-peptide, sequence HLNILSTLWKYR) as a new approach to delivering therapeutics to the CNS, as illustrated by the efficient delivery of small molecules into the CNS after intramuscular injection. This synthetic analog could thus also be used as a vector [28].

#### 3.1.2. Intrinsic Neuronal Protection

In addition to its carrier properties, TTFC has also been reported to display neuroprotective activity. Indeed, retrograde and trans-synaptic transport of TTFC into the CNS after muscular injection is similar to pathways followed by trophic factors [29]. Different studies have demonstrated that TTFC had a trophic action in the brain and that it mimicked growth factors involved in survival pathways [30]. Furthermore, TTFC prevents cell death by apoptosis via several routes. Briefly, apoptosis occurs after cytochrome c release and caspase-3 activation following induction by the apoptotic promoter Bax. This promoter, in turn, is activated upon reversion of Bcl-Xl activity (an anti-apoptotic protein) through the dephosphorylation of the *Bad* gene [31]. Caspases are involved in major apoptotic events such as DNA digestion, chromatin condensation or membrane blebbing. TTFC disturbs these events by inhibiting caspase-3 activity, thus preventing apoptotic cell death [32]. TTFC also protects cells from apoptotic death by inhibiting Bad dissociation and association with Bcl-Xl. TTFC was thus shown to exert anti-apoptotic effects by inhibiting cell death pathways activated by Bad and by blocking pro-caspase-3 activation [33].

Moreover, TTFC is also involved in the activation of survival signaling pathways. Trk receptors transmit their signals from axons to neuronal cell bodies in a retrograde manner. Neurotrophins binding to these receptors allow protein kinase activation and tyrosine residue auto-phosphorylation. These phosphorylated tyrosine residues are then recognized by intracellular signaling proteins activating a kinase cascade. For example, Trk activation can be the trigger for phosphoinositide 3-kinase (PI3K) pathway activation, which in turn activates protein kinase B (also known as Akt), which is necessary and sufficient for eukaryotic cell survival [34]. TTFC stimulates signaling pathways Akt/PI3K, ERK and Raf/MEK/ERK [30,32,34], which play a role in cell survival. For example, PI3K is involved in the prevention of low potassium-induced apoptotic death. TTFC can provide cell protection through its action on survival signaling pathways. TTFC also protects cells by activating p21Ras protein, which is essential for the inhibition of cell death pathways [31,32]. In addition, TTFC treatment affects calcium-related gene expression, suggesting that TTFC can impact anti-apoptotic pathways through calcium-related mechanisms [29].

Another type of protection provided by TTFC concerns oxidative stress. Oxidative damage is observed in neurodegenerative diseases such as Parkinson’s disease (PD). Generation of reactive oxygen species can also contribute to MPP(+)-induced oxidative stress that activates a series of cellular factors that initiate apoptotic cell death [31]. Cubí et al. showed that after the administration of TTFC, the ceramide content and the nSMase activity increased in cerebellar granule neurons and NGF-differentiated PC12 cells [35]. It is well-known that an increase in ceramide level is related to neuron survival. Hence, as TTFC treatment increases ceramide contents, this provides neurons with a better protection against oxidation. TTFC is thus able to protect cells against neuronal oxidative stress and prevent oxidative damage.

Owing to its neuroprotective properties, TTFC has been widely tested in animal models of PD, cerebral ischemia, amyotrophic lateral sclerosis (ALS) or animals treated with molecules inducing apoptosis (Table 2, presented in Section 3.1.3). TTFC was particularly reported to induce protection against dopamine (DA) loss and improved MN behavior [30]. DA plays a key role in controlling locomotion and is the main neurotransmitter affected in PD. DA receptors are involved in the activation of Trk receptors, and it seems that the trkB pathway may be crucial for DA neuron survival. As mentioned above, TTFC has a neuroprotective effect, and that may be due to its known contribution to the activation of Trk receptors, thus improving mechanisms of DA neurotransmission [33]. TTFC had also been identified as a neuroprotective agent in ALS animal models as it prevents anti-apoptotic effects [30]. TTFC fusion proteins were tested in mice with ALS disease and displayed a combination of TTFC neuroprotective and carrier properties. Improvements in motor function activity were observed in TTFC fusion-treated mice, curbing disease progression and increasing the number of surviving MN [29]. In parallel to MN survival, a reduction in microglial reactivity was detected, stopping disease progression. Another example worth mentioning was the use of naked DNA encoding TTFC (nDNA-TTFC) in the treatment of cerebral ischemia. This disease is characterized by a decrease in blood supply to the brain. This dysfunction can lead to permanent disability and death. There is currently no treatment to counter these serious effects. The efficacy of the treatment partly relies on the ability of the therapeutic treatment to reach the neurons after crossing the BBB. Every oxidative stress parameter was reduced after nDNA-TTFC treatment in all brains tested, protecting them from oxidative damage [36]. This treatment may thus represent a non-invasive and non-viral therapeutic approach to treating cerebral ischemia. The unique properties of TTFC as a carrier and neuroprotector may pave the way for many applications to cure neurodegenerative diseases.

#### 3.1.3. Overview of the Uses of TTFC for Its Neurological Properties

Based on all its properties, TTFC has already been used under different forms (e.g., alone, in fusion protein, conjugated) for neuronal applications, including in vivo in several studies, confirming its neurological properties. Examples of these applications are presented in Table 2.

### 3.2. Immunological Properties

#### 3.2.1. Immunological Properties against Tetanus

Having presented the neurological properties of TTFC, we will now review its immunological properties. TTFC contains four universal epitopes of the 11 present in the TT sequence: 52 to 68 [52], 83 to 103 [52,53,54,55,56], 290 to 309/310 to 325 [57] (these two peptide sequences may represent the same epitope) and 409 to 420 [52] in the TTFC sequence. An immuno-informatics analysis performed by Nezafat et al. [58] unveiled four regions of the TTFC as helper epitopes (53–69, 84–108, 220–247, 361–386).

According to other studies, TTFC also contains 13 of the 28 epitopes recognized by more than 75% of patients [52,53,54,55,56,57,59]. This means that nearly half of TT epitopes are contained in the TTFC sequence. Ghafari et al. [60] suggested that TTFC may be the immunodominant part of the toxin, i.e., that the fragment may elicit the humoral immune response. Moreover, TT neutralizing antibodies were shown to be mainly directed against TTFC in mouse and human models (100% for mouse models and 75% in a human model) [61]. It is important to note that the many neutralizing antibodies described since the 1980s [62,63,64] cannot individually neutralize TT in vivo. The development of an active and safe monoclonal antibody (mAb) remains a prospect for upcoming years [65]. Hence, TTFC has become a very interesting candidate from an immunological point of view, as it could protect against tetanus. Ghotloo et al. [66] gathered the state of knowledge on epitope mapping of TT and, more particularly, the epitopes localized on the TTFC by mAbs.

Keeping its immunological properties in mind, TTFC was then tested in vivo against tetanus. The first study conducted by Fairweather et al. tested the last 121 amino acids of the B fragment associated with TTFC in mice. One microgram of the fragment was injected without adjuvant and provided complete protection against TT, as all mice survived the TT challenge [67]. Other studies were conducted on TTFC alone showing that it could induce a 13-month protection against TT when expressed in bacteria, yeasts, plant cells or insect cells [68].

Further studies then compared its immunogenicity to the antigen present in the current vaccine (tetanus toxoid, TTd) [69,70]. Depending on the injection mode, the antibody response changed. No difference between TTd and TTFC was observed after the first transcutaneous delivery (injection of either 30 µg of each protein or molar equivalent doses). However, after the third injection, TTFC induced a 2 to 12 times higher antibody response (mainly IgG1 subclasses responses, IgA were not detected). Kinetics of antibody response of both proteins also seemed to differ. Using TTFC, a constant increase in protective response was observed, whereas using TTd, the level of protective response remained constant event after new injections. Using subcutaneous injection, contradictory results were obtained. In one study, a 100 times higher dose of TTd was required to induce the same protection as TTFC [69], and in another, TTFC was 100 times less potent than TTd [70]. The origin of these differences has not been fully investigated, but the induced protection seems to be linked to the mode of injection, which must be considered when developing new tetanus vaccines. Other modes of administration confirmed the capacity of TTFC to offer protection against tetanus: sublingual immunization using *Bacillus subtilis* as an antigen delivery system showed full protection in mice [71]. The capacity of TTFC to be used as an antigen in a tetanus vaccine has been demonstrated, but further investigations on the impact of the mode of injection must be conducted.

Because TTFC contains many epitopes, its use as a carrier to deliver molecules to the CNS may be delicate in vaccinated individuals. Studies have been conducted to analyze the immunogenic side effects of the administration of TTFC in vaccinated animals [72]. They revealed that the amount of injected TTFC as a carrier (molar equivalent) generally exceeds 10,000 times the lethal dose of TT in a mouse model, and neutralization of all the TTFC injected was unlikely even for a vaccinated animal. This allowed a small part of TTFC to be internalized and to reach the MN. Results have shown that vaccinated and unvaccinated animals presented detectable retrograde transport of TTFC using fluorimetry [72]. This demonstrated that at a high dose, TTFC could be used as a carrier for delivery of bioactive molecules into CNS even in vaccinated individuals. Another explanation could be linked to the mode of injection, as after an intramuscular injection, TTFC was not blocked by antibodies developed after immunization against TT. This could be due to the rapid uptake of the protein after intramuscular injection [21].

Finally, as an attractive alternative to commercial ELISA assays, dipstick tests were developed using gold-conjugated TTFC [73]. This conjugate was used to bind anti-tetanus antibodies in whole blood and plasma samples, with no cold chain requirement. The results obtained were highly promising, with a specificity greater than 0.9. The use of this device could thus be integrated into monitoring programs for populations at risk (non-immune or low titer individuals).

#### 3.2.2. TTFC as a Fusion Protein: Enhancement of Immunogenicity

TTFC did not only induce protection against tetanus, it also enhanced immunogenicity of its partner when fused. DNA-based vaccines have shown their efficacy against infectious diseases but revealed disappointing results when used in tumor models. To overcome this problem, fusion with antigen helpers was developed. TTFC displays functional characteristics of helper antigens and was thus tested in several DNA fusion proteins [74]. In particular, it improved T-cell immunity of viral oncogenes HPV-16 E6 and E7 (viral oncogenes targeted by tumor suppressor proteins p53 and pRb) used in DNA vaccines to protect against papillomavirus [75]. The fusion with TTFC enhanced CD8+ T cell response against the protein of interest. Indeed, the mounting of a robust E7-specific T-cell response was observed after the administration of a TTFC fusion protein [76]. This induced tumor regressing and prolonged survival of all mice tested. E7 stability was also improved by the fusion protein, as E7 accumulation was higher in cells treated with the fusion. This could be explained as TTFC contains universal epitopes for human CD4+ T lymphocytes: the CD4+ T-cell helper response produced after epitope recognition may promote CD8+ T-cell responses. In addition, TTFC has a stimulatory effect on IFN-γ and CD69 production, both involved in modulating the immune response. Enhancement of the immunogenicity using TTFC fusion proteins may thus be due to the action of these proteins [60].

The fragment was also shown to enhance bacterial polysaccharide antigenicity in conjugate vaccines. As for TTd, TTFC has been tested as a carrier protein in conjugate vaccines, albeit because the production of TTFC does not require a formaldehyde detoxification step, its conjugation sites are easier to characterize [77]. Several authors chose to use this protein to synthetize well-defined glycoconjugates to highlight the link between the conjugate structure and the induced immune response [78,79,80,81]. TTFC was shown to play a normal protein carrier role as its conjugation with the bacterial polysaccharides induced a higher and long-lasting immune response.

#### 3.2.3. In Silico Design of Epitope-Based Vaccines

In the last decade, in silico methods have been used to design vaccines based on TTFC epitopes associated with other epitopes. For example, in the case of atherosclerosis, a multi-epitope construct was designed by fusing epitopes from TTFC with other immunogenic molecules such as calreticulin, heat shock protein 60, and cholera toxin B [82]. Bioinformatics analysis allowed the authors to build a stable chimeric protein with the potential to shift the immune response and to reduce atherosclerosis. In silico approaches were also used to design anti-infective vaccines against brucellosis [83] and leptospirosis [84] in order to induce a strong immune response mediated by T- and B-cells. Immuno-informatics was also helpful to design a vaccine against melanoma [85], by fusing several antigens, including TTFC. This multi-epitope vaccine (MEV) approach showed high immunogenicity, providing hope for cancer immunotherapy. In another study [58], different algorithms and servers were used to design a novel MEV against cancer. The selected cytolytic T lymphocyte (CTL) epitopes were linked together to enhance epitope presentation. In parallel, different helper epitopes (including TTFC) were conjugated to stimulate helper T lymphocyte (HTL) immunity. All these epitopes were associated with heparin-binding hemagglutinin used as an adjuvant. The final protein was thus able to stimulate both cellular and humoral immune responses. An in vivo study was conducted to produce the corresponding vaccine in *E. coli* in order to evaluate antitumor efficacy against the HPV-16 E7-expressing murine tumor cell line TC-1. The results obtained revealed a significantly higher IgG secretion with MEV containing TTFC epitopes compared to the E7 protein vaccine [86].

#### 3.2.4. TTFC Uses for Its Immunological Properties

As for its neurological properties, the immunological properties of TTFC have inspired many groups of research. Again, TTFC was used under different forms (e.g., alone, in fusion proteins, in conjugate vaccines) for diverse vaccine applications. Some examples of these applications are presented in Table 3.

## 4. TTFC Production

Since the discovery of TTFC, different methods of production have been developed (Figure 4). The first studies were conducted using papain digestion of TT and, as a consequence, TTFC was first obtained using this technique. Nevertheless, the hazardous character of this manipulation has led researchers to develop recombinant systems to express the protein. Bacterial recombinant hosts are currently the most widely used to produce the protein, but other systems such as yeasts, plant cells and insect cells have also been used. In this part, we describe the most common methods to produce TTFC and provide information to enable the comparison of the production methods described.

### 4.1. Papain Digestion

One of the simplest ways of obtaining TTFC is to enzymatically digest TT. This can be achieved using the protease papain present in papaya fruits. Papain cleaves TT between Ser864 and Lys865 residues resulting in two fragments: TTFC and fragment A-B (Figure 1). Papain was first used for this purpose by Helting et al. in 1974 [7]. At that time, structural and functional studies of TT were difficult to ascertain because of the lability of the whole protein after purification. This is why researchers decided to split the protein into smaller fragments, trying to obtain separate structural information. The digestion was usually performed in 0.1 M phosphate buffer containing 1 mM EDTA and cysteine hydrochloride at pH 6.5. The experimental conditions (e.g., reaction temperature, amount of material) were progressively optimized [7,100,101,102,103]. For example, in 2014, Murzello et al. [100] digested TT within 30 min at 55 °C using 10 units of papain from papaya latex (Sigma P4762) per 10 mg of TT. To stop the reaction, the enzyme was inactivated by adding L-1-chloro-3-tosylamido-7-amino-2-heptanone and by letting the batch cool to 25 °C.

In the next step, TTFC was purified from the solution obtained after papain digestion. Usually, several chromatographic separations (e.g., size-exclusion chromatography) were performed to finally yield pure TTFC. Subsequently, an affinity chromatography column using an anti-fragment [A-B] antibody could complete the purification protocol. Sometimes, TTFC needed to be rechromatographed (on the same column material under different elution conditions) to remove the remaining contaminants. Yields obtained after all purification steps varied between 15 to 28% from the TT starting material (approximately 0.13 to 10 mg of protein per mL) [7,100,101,102,103].

This method is easy to handle and was mostly used at the beginning of TTFC discovery. Since then, different recombinant systems have been employed to produce TTFC without the use of a toxic protein as a starting material.

### 4.2. TTFC Production in Recombinant Systems

TTFC production using papain digestion was the first to be described, but as it was hazardous and a major difficulty was to eliminate the very small amount of undigested toxin from the fragment preparation, recombinant production systems were developed [104]. These systems present some striking advantages compared to papain digestion. First, large amounts of TTFC can be efficiently produced since large numbers of host organisms (e.g., bacteria, yeast, plant cells, insect cells) can be grown in small- and large-scale cultures with standard laboratory equipment. Second, only controlled non-toxic proteins can be produced, which means that production is safer and toxic contaminations are easy to avoid [105].

#### 4.2.1. *Escherichia coli* as a Host for TTFC Production

In 1986, the cloning of the DNA encoding the TTFC from *C. tetani* CN3911 strain was published [106]. From this first study, several groups started to work on the production of TTFC as a heterologous protein in *E. coli* (Table 4) [104,105,106,107,108,109,110,111,112,113]. Low amounts (from 1 to 10 mg/L) of recombinant proteins were obtained from expression vectors containing *trp* or *tac* promoters [104,105,106]. TTFC was also produced as a fusion protein in the pMalc2x vector [109]. TFC codon analysis revealed that the gene contained a high proportion of rare codons in *E. coli*, and teams thus produced synthetic TTFC DNA in order to reduce rare codons and decrease the percentage of AT pairs, increasing protein production [110,111]. However, it appeared that the yield increased further when BL21 bacteria and an expression plasmid the including T7 promotor were used [107,108,112,113]. In 2011, Yu et al. [110] optimized the fermentation production of 40 L to reach 333 mg/mL recombinant TTFC. As described recently [114], the fermentation conditions were further optimized to increase the production of recombinant proteins. In the late 1990s, other bacterial expression systems were studied to produce and ensure the delivery of TTFC (Figure 5). In fact, these alternative bacterial expression systems were developed to produce oral vaccines against TT, i.e., they served both as an expression method and as a delivery system.

#### 4.2.2. TTFC Expression and Delivery in Other Bacterial Host Strains

The production of heterologous proteins in cyanobacteria, phototrophic microorganisms with low nutrient requirements, may be an alternative. For eukaryote proteins, this is possible in the form of fusion proteins. Recently, in 2021, the TTFC protein was expressed in the cyanobacterium *Synechocystic* sp. PCC6803 (*Synechocystis*) [115]. This bacterium was initially modified to stably express the tobacco etch virus protease (TEVp). Then in the *cpc* operon locus, the native *cpcB* gene encoding the abundant β-subunit of the phycocyanin was replaced with a fusion construct comprising the *cpcB* and TTFC DNA sequences separated by a TEV cleavage site. In vivo cleavage led to the accumulation in the cytosol of soluble TTFC proteins and putative aggregates containing the uncleaved protein cpcB-tev-TTFC, the sum of the two proteins representing approximately 20% of the total protein content (TPC). Hidalgo Martinez et al. [116] showed that the accumulation of soluble cpcB-tev-TTFC fusion proteins could occur as they contribute to the formation of phycobilisomes, comprising the major light-harvesting antenna complex for photosynthesis.

TTFC was produced by other bacterial recombinant expression systems for different applications. All these bacteria were not only used to express TTFC but also to deliver it in oral vaccination approaches. These delivery systems were developed in order to produce safe, stable and inexpensive oral vaccines. Such vaccines are particularly appropriate in developing countries because syringes and needles used for injected vaccination may be used repeatedly and lead to the transmission of infectious agents such as HIV, hepatitis B and C viruses. Oral immunization also simplifies the administration of vaccines, allowing less qualified health workers to immunize populations. These vaccines are highly compatible with mass immunization programs as their logistics are simpler. Finally, oral administration is usually preferred by both children and adults compared to parenteral injection [117]. The objective of oral vaccines is to pass through the intestinal mucosal surfaces to reach the immune system. Mucosal surfaces play a role in nutrient uptake, and an immune response against these dietary antigens is unwanted and generally suppressed to avoid food intolerance. It is thus difficult to induce a systemic immune response after the delivery of antigens to these surfaces. The immune response can, however, be improved by associating the antigen with a bacterium [118]. Lactic acid bacteria are well adapted to deliver antigens through mucosal surfaces and are interesting since they are generally recognized as safe (GRAS) by the FDA [119]. These nonpathogenic food-grade Gram-positive bacteria can be resistant to the harsh conditions of the intestinal environment. For example, the *Lactococcus lactis* bacterium can prevent direct contact of antigens with gastric acid and proteolytic enzymes, conferring a higher resistance in the intestinal medium [120]. Different lactic acid bacteria were tested as expression systems for TTFC: *L. plantarum*, *L. lactis* and *L. casei* [118,119,120,121,122,123,124,125,126,127]. Generally, the recombinant plasmids were constructed in the *E. coli* DH5α host strain and were subsequently introduced into alternative bacterial host strains by electroporation. Several strains and plasmids were tested to express the protein, leading in most cases to successful intragastric immunization. Strains UCP1060 and MG1363 were most often used for *L. lactis*, strains NCIMB8826 for *L. plantarum* and strains ATCC393 for *L. casei* [118,119,120,121,122,123]. Plasmids pMEC46 and pMEC127 were both frequently used within these strains, and the administration of MG1363(pMEC46) and NCIMB8826(pMEC127) strains, in particular, showed a high antibody production in mice [119,121,123]. To increase the immune response, different mutations were also tested in bacterial strains. For example, *L. plantarum* and *L. lactis* (Alr^−^) mutant strains (replacing of L-alanine by D-alanine), both producing TTFC intracellularly, were more immunogenic than their wild-type counterparts [121]. These expression systems were good alternatives to attenuate pathogenic bacterial delivery systems such as *Salmonella typhimurium* [128] and *Mycobacterium bovis* BCG [129].

For TTFC oral delivery, *Streptococcus gordonii* and *Bacillus subtilis* spores have also been used. *S. gordonii* is a nonpathogenic Gram-positive commensal bacterium, which is able to colonize mucosal surfaces [130]. This bears the huge advantage of continuously stimulating the immune system because such bacteria act on dendritic cell maturation [131]. Dendritic cells are linked to the induction of primary T-cell responses and undergo a maturation process to provoke such immune responses. This maturation is launched by inflammatory signals and ends upon contact with T-cells. During their maturation process, dendritic cells migrate from peripheral tissues to lymphoid organs and exhibit a strong antigen-presenting capacity. Due to their central role in the immune response, dendritic cells have been targeted for vaccine development [131]. *S. gordonii* appeared as a good candidate for antigen delivery in this context because it was shown to induce the maturation of dendritic cells and thus elicit an immune response. The GP1253 strain was used in two studies dealing with the production of TTFC in *S. gordonii* using either plasmid pSMB55 or pMSB158 [130,131]. The production resulted in the expression of TTFC in 78% of recombinant cells (100 ng/10^9^ CFU of bacteria) and the administration of these recombinant bacteria protected mice against the TT challenge [130]. More recently, *B. subtilis* spores were investigated as an antigen delivery system (Figure 5). This strategy bears several advantages: spores are considered to be safe for human and animal use, and they can stimulate cytokine release and interact with antigen-presenting cells [132]. Moreover, their large-scale production is inexpensive, the genome of *B. subtilis* is known and is accessible for genetic manipulations. Spores also exhibit stability towards solvent exposure and extreme temperatures, which enables them to maintain integrity in extreme environments, including in the intestine [132]. TTFC was produced in all cases as a fusion with a spore coating protein (CotC or CotB). This fusion led to the expression of TTFC on the *B. subtilis* spore surface. Different plasmids and strains were applied, and in each case, a TTFC-specific antibody response was obtained [133,134,135]. More recently, the spore-display strategy was optimized by altering the spore coating protein and spore production temperature [136].

In other studies, attenuated *S. typhimurium* and *Bordetella bronchiseptica* Gram-negative bacteria were used for TTFC expression. These recombinant systems were developed as live bacterial vector vaccines to induce double protection within the same vaccine: one directed against the tetanus toxin and the other against the host bacterium. Both bacterial strains were used for oral vaccine development. *B. bronchiseptica* infects the respiratory tract and hence, could be a suitable vector for oral immunization. A plasmid encoding the TTFC (pFHAFrgC) transformed into the BBC18 strain was investigated. All immunized mice developed a protective immune response against the TT challenge and displayed high antibody titers against *B. bronchiseptica* [137]. Attenuated *Salmonella* strains used to express TTFC were used before as vaccination strains to raise an immune response against the *Salmonella* pathogen itself and appeared to be safe and efficient [138]. Different foreign antigens were successfully expressed in *S. typhimurium,* but their capacity to induce an immune response was sometimes compromised because of the instability of their plasmid. This relation between plasmid instability and the generated immune response directed against the foreign antigen was explored by different research teams. For example, Dustan et al. [138] compared the stability of different plasmids expressing TTFC in the attenuated BRD509 strain of *S. typhimurium* (*aroA aroD* mutant). In all cases, optimal antibody response was generated against *S. typhimurium*, but differences were observed concerning the anti-TTd antibody response. They showed that by using the unstable plasmid pIC20H/TT, no anti-TTd antibody was detected in mice, whereas by using the highly stable plasmid pTET*tac4*, the antibody titers were significantly higher. A third plasmid, pACYC184/TT, presenting intermediate stability, gave rise to an intermediate tetanus vaccine efficacy. This corroborates the link between the plasmid stability and the quality of the protection directed against the expressed antigen [138]. The stability of foreign antigen expression can be improved by using inducible promoter systems such as *nirB*, *dmsA* or *tac* [138,139,140]. In each case, the use of the promoter stabilized TTFC expression and induced higher TT antibody titers in immunized mice. Among these promoters, it was demonstrated that the pTET*dmsA*3 plasmid expressed in *S. typhimurium* generated higher titers than the pTET*nir*15 plasmid, expressed in the strain CVD908-*htr*A [139]. Another study conducted in the BRD509 strain revealed that the pTET*nir*15 plasmid induced stronger protection than plasmids containing the *tac* promoter [140]. In addition to the plasmids, the nature of the strain used as the live vector also influences the quality of the immune response directed against the foreign antigen. A study was conducted on two types of *S. typhimurium* mutant strains using TTFC-expressing plasmids: BRD509 and BRD807 (*aroA htrA* mutant). Independently of the tested plasmids, anti-TT antibody titers were always superior when mice were immunized with the BRD509 strain [141].

In summary, several bacterial production systems have been developed to produce TTFC. Different modifications have been made to the plasmids and the host strains to improve the fragment expression. Bacterial expression systems are widely used because they are cheap, their genetic manipulation is easy, and these fermentation systems are often easy to scale up. However, expressing foreign antigens in bacterial strains also has its drawbacks: Gram-negative bacteria cell walls contain lipopolysaccharides (LPS), which are toxic pyrogens. Consequently, proteins expressed in these bacteria must be carefully tested on LPS content before being used. Yeasts or plant cells exhibit no such features and could thus be attractive hosts for TTFC expression.

#### 4.2.3. Recombinant TTFC Expression in Yeast and Plant Cells

The yeast species *Saccharomyces cerevisiae* and *Pichia pastoris* were used for recombinant TTFC expression as well as tobacco chloroplasts, representing a common plant expression system. To obtain satisfying levels of expression in yeast, a synthetic TTFC gene was constructed, containing a codon-optimized high CG gene lacking the fortuitous polyadenylation sites, which gave rise to truncated mRNA [142,143]. Around 60 to 90 mg/L of soluble protein were obtained using *S. cerevisiae* S150-2B strain expressing the synthetic TTFC gene (pWYG5-TET15) [142]. However, in *S. cerevisiae*, the recombinant TTFC secreted in the culture medium was immunogenically inactive due to glycosylation. A methylotrophic yeast, *P. pastoris*, produced a much higher yield of 12 g/L of soluble protein using the integrative plasmid with the promotor from the methanol-induced alcohol oxidase gene and the synthetic TTFC gene (pPIC3-TET15) [143]. When the native gene was cloned into a yeast expression plasmid (p9k-G) with the alpha factor secretion signal and expressed in *P. pastoris,* the secreted TTFC was glycosylated. Site-directed mutation performed on five potential N-glycosylation sites showed that the number and localization of these sites impact the expression and secretion of TTFC. Indeed, decreasing the number of N-glycosylation sites decreased TTFC secretion [144].

Conversely, plant cells did not require the use of the synthetic TTFC gene to obtain high yields of proteins. Indeed, the AT-rich bacterial gene, when expressed in a cassette consisting of a PrrnLT7g10 cassette (plasmids pJST10 and pJST11, respectively), produced twice as much protein (25% of total cell protein (TCP)) than the high-GC synthetic gene. However, this high level of TTFC is detrimental to plants, as evidenced by the chlorotic phenotype in plants [145,146]. The chloroplast expression system has inherited similarities with the prokaryotic expression systems derived from the ancestral cyanobacterium. They contain a plastid-encoded RNA polymerase, which can be compared to the bacterial RNA polymerase [147]. Two types of methods can be used to produce proteins in plant expression systems: either nuclear transgenic plants (low expression levels generally achieved) or DNA can be introduced into the chloroplast genome by particle bombardment (higher expression levels of recombinant proteins) [148]. Proteins are usually extracted from leaf tissues using an extraction buffer. In addition, chloroplasts are heat stable, cheap to produce, and they can remove undesirable selectable markers and use operons for multi-antigen expression; they produce a high level of immunogenic recombinant proteins, specifically target the gene and retain the gene product in the plastid; this system also benefits from environmental containment as a result of maternal inheritance [148]. Michoux et al. studied the production capacity of the *Nt*-pJST12 line obtained from a plasmid expressing the synthetic gene in a cassette consisting of a PrrnLatpB cassette. They showed that 92 g of TTFC could be produced annually, using five 250 L-bioreactors and running 20 rounds of production per reactor per year. This quantity of TTFC should be enough to launch preclinical and clinical trials [149]. Chloroplast-expressing systems were able to yield high amounts of protein but were not always efficient as vaccines. As both plant and the expressed vaccine are degraded in the intestinal tract, oral immunization was inefficient [150]. Nevertheless, intranasal immunization was protective in mice against the TT challenge [145,146].

Beyond yeast and chloroplasts, insect cells were also tested as host organisms using baculovirus expression vector p36C to co-infect *Spodoptera frugipedra* cells. The resulting TTFC was soluble and showed an immunogenic activity, but its secondary structure was different from the TTFC obtained in *E. coli* as its ganglioside binding properties were altered [151].

TTFC can be produced using a panel of different production methods. Purification [104,105,107,108,110,152] and characterization [108,110,122,133,152] steps are then usually the last ones performed in protein production to finally obtain the required quality. In 2020, Chai et al. established a simple method to purify TTFC by ion-exchange chromatography [153].

## 5. Conclusions

The development of new vaccines and neuroprotective agents present major issues in drug discovery. New vaccines must be well defined and better controlled, present great antigenic and immunogenic activities and be as stable as possible to allow mass vaccination campaigns in developing countries. The development of neuroprotective agents mostly focuses on the drug delivery into neuronal cells as the BBB is difficult to cross. TTFC protein can be a good candidate to overcome problems in both fields. Its structure has been characterized, allowing a better understanding of its biological properties. Recombinant TTFC is easily characterized and was shown to be sufficient for the protection of mice against tetanus. Its properties can also be used to protect against other diseases (e.g., multiple myeloma, cholera) as TTFC enhances the immunogenicity of therapeutic proteins when fused to them. This fragment is responsible for the neuronal binding of TT, meaning that it can reach neurons. This property has been widely studied for drug delivery into the CNS, as this is a key point in the cure of neurodegenerative diseases (e.g., ALS, AD, PD). These studies have shown that when conjugated or fused to a drug, TTFC retains its neuronal binding capacity and is a good carrier for drug delivery into the brain. Intrinsic neuroprotective properties were also discovered, confirming the growing interest for this protein in the neurological field. TTFC thus appears to be a promising protein for drug discovery, as its production, characterization, and properties are well described. Further studies must now be conducted on different animal models to confirm its biological relevance and hopefully will result in the development of new drugs. In addition, TTFC-based immune-informatics approaches also seem to be an attractive and effective design to access new therapeutic entities against certain zoonotic diseases (e.g., brucellosis, leptospirosis).

In conclusion, TTFC protein can be considered to be a useful drug discovery toolbox. Few proteins offer such a wide range of uses, including as a carrier, delivery vector, conjugate, booster, inducer, and neuroprotector. With the development of nanomedicines, and consideration for brain safety [154], the use of TTFC could pave the way for novel, safer applications.

## Figures and Tables

**Figure 1 pharmaceuticals-15-00756-f001:**
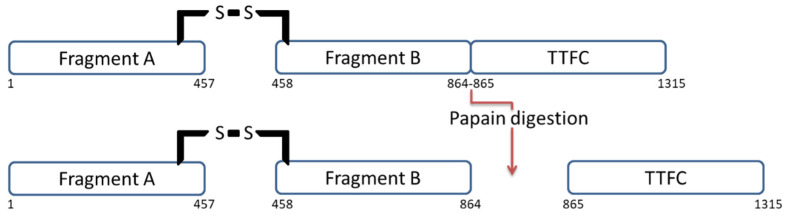
TT structure before and after papain digestion. Papain digests the protein by splitting it into two fragments: TTFC and fragment A-B.

**Figure 2 pharmaceuticals-15-00756-f002:**
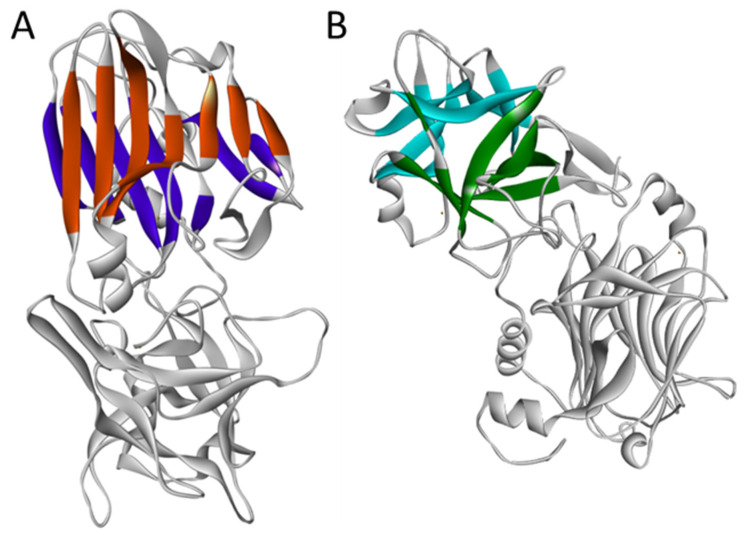
TTFC jelly roll (**A**) and β-trefoil (**B**) domains. (**A**): in dark orange, concave solvent-exposed face; in dark purple, convex solvent-exposed face. (**B**): in blue, β-hairpin triplet; in green, β-barrel. Picture obtained after reprocessing PBD files 1A8D and 1AF9 using Discovery Studio^®^ software (BIOVIA Dassault Systèmes, Discovery Studio, Release 2017, San Diego, CA, USA).

**Figure 3 pharmaceuticals-15-00756-f003:**
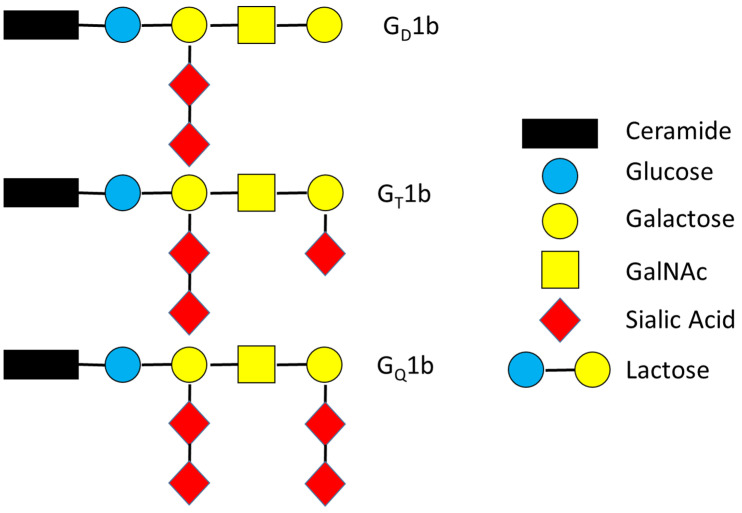
Structures of gangliosides G1b.

**Figure 4 pharmaceuticals-15-00756-f004:**
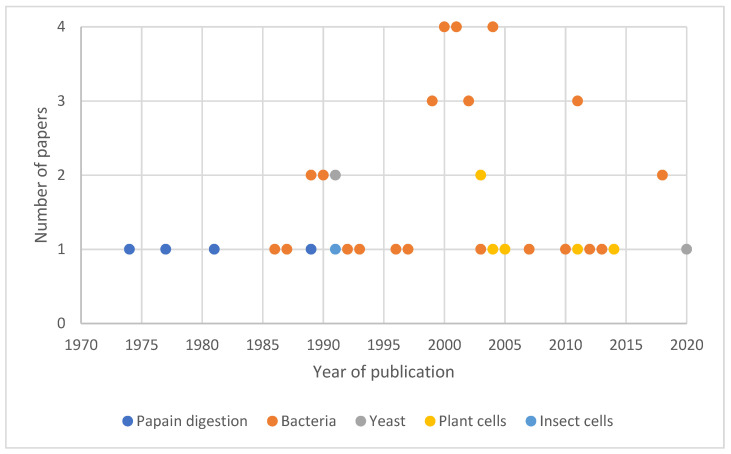
Diversity of TTFC production over the years. Before 1984, only papain was used to obtain TTFC (in blue); then recombinant systems were developed, the most widely used being bacteria (in orange). Some studies were also conducted on TTFC production in yeast (in grey), plant cells (in yellow) and insect cells (in light blue). The number of papers published using these methods are provided on the *Y*-axis.

**Figure 5 pharmaceuticals-15-00756-f005:**
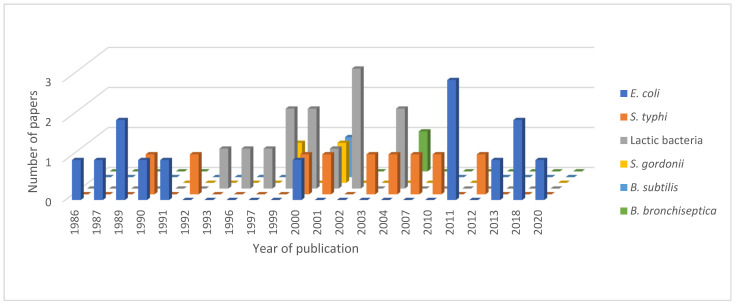
Distribution of the expression systems for bacterial production of TTFC over the years. *Escherichia coli* (in blue) was the first recombinant system used to produce TTFC; between 1990 and 2012, *Salmonella typhi* (in orange) and lactic acid bacteria (in grey) were the most studied system of expression. Other bacteria were used less often as expression systems for TTFC (*Streptococcus gordonii* in yellow, *Bacillus subtilis* in light blue and *Bordetella bronchiseptica* in green).

**Table 1 pharmaceuticals-15-00756-t001:** Terms and abbreviations used in association with tetanus toxin fragment C.

Key Terms Used	Abbreviations
Tetanus toxin fragment C	TTFC or TTFrC or TtxFC
Tetanus toxin C fragment	TTFC or TTC or TetC or TCF or TTCF
Tetanus toxin heavy C fragment Non-toxic carboxylic fragment of tetanus toxin	TeTx H_c_ or H_c_-TeTx
Recombinant tetanus toxin fragment C	rTT-Hc
Tetanus toxin native heavy C-fragment	TeNT-Hc
Heavy C fragment wild-type	HcWT
Carboxylic fragment of tetanus toxin	HC
C-terminal fragment of tetanus toxin Fragment C ot tetanus toxin	FrgC or FrC

**Table 2 pharmaceuticals-15-00756-t002:** Non-exhaustive list of in vivo uses of TTFC for neurological applications between 2005 and 2022.

Medicinal Product	Biological Interest	Administration and Dose	Experimental Model	Observed Effects	Ref.
** *TTFC used alone* **					
TTFC	neuronal protection (ALS)	Intramuscular 1 μg	male and female SOD1-G93A mice	Modulated the levels of NLRP3 and caspase-1 in spinal cord, EDL and SOL muscles Reduced IL-6 levels in tissues drastically affected by ALS Potential therapeutic molecule	[37]
TTFC	neuropsychiatric disorders (depression)	intramuscular 20–60 μg/kg	adult male Wistar-Kyoto rats	Levels of hippocampal and frontal cortical BDNF increased Levels of TNF-alpha in the same areas decreased Potential utility of TTFC in PD-depression comorbidity	[38]
TTFC	neuronal protection (spinal MN degeneration)	direct spinal infusion (total amount of ~42 ng/rat) intramuscular (total amount of ~400 ng/rat)	adult male Wistar rats	Attenuated the AMPA-induced astrogliosis Increased the phosphorylation of the TrkA receptor at Y490 in spinal MNs Intramuscular > spinal infusion	[39]
TTFC	neuronal protection (PD)	intraperitoneal 0.5 mg/kg	male 8-week-old Sprague–Dawley rats	TTFC as pre-treatment Prevented decrease in DA, TH, DAT, VMAT-2 Uses in neuronal dysfunctions	[40]
TTFC	neuronal protection (AD, effect on learning and memory)	medial septum (local administration) 100 ng	adult male Wistar rats	Protection of the cholinergic system After administration of a toxic peptide, TTFC functionally maintained memory Lower level of cell degeneration Maintained cell morphology	[41]
TTFC	neuronal protection (post-methamphetamine treatment)	intramuscular 40 μg/kg	adult male C57BL/6J mice	Only three injections of TTFC Prevented the striatal tyrosine hydroxylase (TH) and DA transporter (DAT) decrease induced by METH Potential for TTFC use against the damage induced by METH	[42]
TTFC	neuronal protection (restorative effect)	intramuscular 20 µg/kg	adult male Wistar rats	Neurodegeneration caused by 6-OHDA Prevented the progression of asymmetrical motor behavior Decreased the neurodegenerative process (fewer dark cells) Decreased of striatal neurodegeneration after 28 days	[43]
Naked DNA encoding for TTFC	neuronal protection (cerebral ischemia)	intramuscular 200 µg	adult male Mongolian gerbils	Improved neurological status and survival Elimination of ischemia-induced motor hyperactivity and oxidative stress Reduced nitrite levels, O_2_-production and lipid peroxidation Improved SOD activity	[36]
Naked DNA encoding for TTFC	neuronal protection (ALS disease)	intramuscular 300 µg	SOD1-G93A mice	Delayed onset of symptoms Extended the mouse survival	[29]
** *TTFC used as a fusion protein* **					
TTFC fused with rAAV8, CMV and eGFP	tracing study (connectivity map)	hippocampal injection 1 µL	adult male and female tdTomatoJ mice	Exploration of the sequence of cerebellar-hippocampal connections Displayed eGFP positive cells in the rhinal cortex and subiculum	[44]
TTFC fused with GDNF	neuronal protection (ALS disease)	intramuscular 300 µg	SODG^93A^ mice	Improved mice survival Delayed onset symptoms Improved motor function Activation of survival signals in SC Treatment by fusion protein was less efficient as GDNF alone	[45]
TTFC fused with GFP	study of neuronal network (study of nerve injury)	/	transgenic mice (NPY-Cre, ZWX)	Allowed to study the consequence of an injury and especially the CNS reorganization circuits	[46]
TTFC fused with IGF-1	neuronal protection(age related nerve alteration)	intramuscular 10 µg	old control FVB and DBA mice	Prevented age-related alterations to nerve terminal at the NMJs Prevented Ca^2+^ dependent contraction No effect with TTFC alone	[47]
TTFC fused with GFP or β-galactosidase	study of neuronal network (muscle specific spinal motor circuitry)	intramuscular 10.57–19.2 µg/mL	new born BalbC/J mice	Fusion protein kept TTFC retrograde transport properties intact With low injected doses, fusion protein spread on other muscles	[48]
TTFC fused with SOD1	neuronal delivery (protein)	intra- cerebroventricular	adult male C57BL6 mice	Enhanced protein distribution and persistence throughout the CNS Injection mode difficult to manage	[24]
** *Other forms of TTFC (analog, complex, conjugate)* **					
^125^I-TTFC	retrograde transport (spinal cord)	intramuscular 10 µg of radiolabeled TTC	transgenic mice (C57BL6, SOD1^93A^)	Quantification of the net retrograde axonal transport Monitoring of a new therapy	[49]
PEISH-based NP with HC	neuronal delivery (gene therapy)	subcutaneous 150 μL of dispersion (conc. 7.5 µg pegylated HC per 2 µg of pDNA)	male 4-month old Wistar rats	PEISH-HC-functionalized NP In day 5, GFP expressed in dorsal root ganglia neurons Gene therapy strategies	[50]
Synthetic analog of TTFC, Tet1-peptide	neuronal delivery (small molecules)	intramuscular 1 µL/g of body weight)	young adult male heterozygous rats	Delivery of small molecules into the CNS without toxicity	[28]
TTFC chemically coupled to GDNF	neuronal delivery (therapeutics)	intramuscular 60–100 µg	adult male mice	The conjugate maintained both TTFC transport and GDNF neuroprotection properties Improved GDNF delivery into MN GDNF persistence in spinal cord section	[51]

**Table 3 pharmaceuticals-15-00756-t003:** Non-exhaustive list of in vivo uses of TTFC for vaccine applications between 2009 and 2020.

Medicinal Product	Biological Interest	Administration and Dose	Experimental Model	Observed Effects	Ref.
** *TTFC used alone* **					
TTFC	tetanus antitoxin	Intramuscular 0.625–15 mg	horses	TTFC was safe and effective for tetanus antitoxin production	[87]
TTFC	vaccine (tetanus)	/	mAbs obtained after BALB/c mice immunization with TT	TTFC 1155–1171 epitope has shown to protect 80% of mice against a lethal dose of TT The antibody response of mice immunized with TT, evaluated with TTFC, showed that anti-TTFC and anti-TT titers were equivalent	[68]
		0.1 mg	BALB/c mice		
TTFC	vaccine (tetanus)	transcutaneous 30 µg	BALB/c mice	TTFC induced higher anti-TT and anti-TTFC antibody titers than the TTd TTFC more immunogenic than TTd	[69]
** *TTFC used as fusion protein* **					
TTFC fused with *S. aureus* coagulase R domain	vaccine (*S. aureus*)	intramuscular 30 µg of TTFC-CoaR	BALB/c mice	TTFC increased immunogenicity of CoaR Higher T-cell response with TTFC-CoaR vaccine than with CoaR alone	[88]
TTFC fused with several epitopes	cancer vaccine (HPV-induced cancer)	subcutaneous 1.5 nmol of MEV (100 µL)	C57BL/6 mice	HTL epitopes (TTFrC and HLA PADRE) and CTL epitopes (WT-1 and HPV E7) Prevention: 100% of immunized mice remained tumor-free Therapeutic: immunized mice had significantly smaller tumors and fewer metastases	[86]
TTFC fused to flagellin	mucosal vaccine (tetanus)	intranasal 2.75 μg	female BALB/c mice	FlaB-TTFC induced strong TLR5 stimulating activity Potential candidate for the development of polyvalent vaccines	[89]
TTFC fused with DNA	cancer vaccine (multiple myeloma)	intramuscular 6 times 1 mg fusion vaccine	clinical trial—phase I 14 patients with multiple myeloma	Idiotype-specific immune response was observed in 29% of patients 43% of patients showed immune response to TTFC alone	[90]
TTFC fused with *Tem 1* cDNA	cancer vaccine (tumor vasculature)	intramuscular 50 µg of plasmid in saline	C57BL/6 and BALB/c mice	Vaccine reduced tumor vasculature compared to control Specifically induced a cellular immune response that controlled tumor progression	[91]
TTFC domain fused with DNA (PSMA_27–35_)	cancer vaccine (prostate)	intramuscular 5 times 400–3200 µg of fusion vaccine	clinical trial—phase I/II 32 HLA-A2^+^ patients and 32 HLA-A2^−^ control patients	Induced DOM CD4+ specific and PSMA27-specific CD8+ T cells After week 24, significant increase in CD4+ and CD8+ specific T cells Safe vaccine which generated anti-PSMA responses in the majority of patients	[92]
TTFC fused with naked DNA (V_H_CDR3_109–116_)	cancer vaccine (lymphoma)	intramuscular 50 µg DNA plasmid	male C3H/HeN mice	Induced immune response Provided strong protective anti-tumor immunity Ensured completed long-term tumor free survival of mice	[93]
TTFC fused with DNA	DNA vaccine (HPV 16 E6 and E7)	intradermal tattoo vaccination 20 µg	C57BL/6 mice	TTFC enhanced the immunogenicity of fused antigens TTFC might promotes CD8+ T cells responses	[76]
TTFC fused with *Cryptosporidium parvum* antigens	vaccine (*Cryptosporidium* *parvum*)	*per os* single dose 5 × 10^9^ CFU	female C57BL/6 and IL18-KO mice	TTFC may play a stabilization role for fusion protein expression Induced specific antigens: IgA detected	[94]
** *Other forms of TTFC (conjugate, bacteria)* **					
TTFC conjugated to pneumococcal polysaccharide	vaccine (Pneumococcus)	intraperitoneal 2 µg/mL of PS per vaccine	female BALB/c mice	Conjugation with TTd, CRM_197_ and TTFC TTFC increased the immunogenicity of the vaccine TTFC is an efficient carrier as those previously used	[95]
TTFC and *S. aureus* surface protein A (SasA)	combined vaccine (tetanus and *S. aureus*)	intraperitoneal 10 µg SasA + 10 µg TTFC	female BALB/c mice	Effective protection against both tetanus and *S. aureus*	[96]
TTFC conjugated to Her2 protein fragment	cancer vaccine (Her2+ breast cancer)	subcutaneous 50 µg of conjugate, 4 boosters of 25 µg	female BALB-neuT mice	50% long-term survival rate with Her2-TTFC vaccine vs. 0% with Her2-only vaccine	[97]
TTFC conjugated to *Burkholderia pseudomallei* PS	vaccine (melioidosis)	intraperitoneal 66 µg of conjugate per dose	female BALB/c mice	The conjugate showed higher levels of IgG than the mix of PS and TTFC Efficient protection against *B. pseudomallei*	[98]
TTFC conjugate to *Vibrio cholerae* OPS	conjugate vaccine (cholera)	intramuscular and intradermal 10 µg of OPS per animal (5:1 conjugate molar ratio OPS:TTFC)	female Swiss- Webster mice	Induced OPS-specific memory response Induced anti-OPS response	[80]
Cytomegalovirus expressing TTFC	vaccine (tetanus)	intraperitoneal 5 × 10^6^ pfu	age-matched female 129S1/SvlmJ/Cr mice	A 13-month protection was induced after a single dose injection	[99]
*Bacillus subtilis* expressing TTFC	vaccine (tetanus)	sublingual and intranasal 1 × 10^9^ cells of died TTFC-expressing *B. subtilis*	weaned piglets	Stimulation of both systemic and mucosal response Equal protection compared to the standard tetanus vaccine	[71]

**Table 4 pharmaceuticals-15-00756-t004:** Expression of recombinant TTFC in *E. coli*.

Expression Conditions	Fairweather et al. 1986 [106]	Makkof et al. 1989 [104]	Makkof et al. 1989 [111]	Halpern et al. 1990 [105]	Ribas et al. 2000 [112]	Motamedi et al. 2011 [109]	Yu et al. 2011 [108]	Yu et al. 2011 [110]	Yousefi et al. 2013 [107]	Aghayipour et al. 2018 [113]
host	DH1	*E. coli*	*E. coli*	DH5α	BL21	DH5α	BL21	BL21	BL21	BL21pLys
TTFC DNA origin	*C. tetani*	synthetic for end of TTFC	synthetic (optimized codons for TTFC)	*C. tetani*	*C. tetani*	*C. tetani*	synthetic	synthetic (optimized AT: 72.50% to 52.47%)	*C. tetani*	*C. tetani*
recombinant protein	TrpE-TTFC (trpE: anthranilate synthetase)	1: met-3AA INFγ-TTFB(537–864)-TTFC(865–1315) 2: met-TTFC	met-TTFC	fusion with 8AA from vector and 9AA from fragment B	112AA Trx-45AA TTFC- His-tag	MBP-TTFC (MBP:maltose binding protein)	Trx-TTFC-6His tag	no tag	Cterm of TTFC (25 kDa)-6His tag	6His-tagged fusion protein
plasmid	pWRL507	pTET-Tact1 pTET-Tact2	pTET-Tact2	pTTQ8	pET32a	pMalc2x	pTIG-Trx	pET32a+	pET28b+	pET28a pET22a
promotor	trpE	tac (derived from trp and lac UV5)	tac	tac	T7	tac	T7	T7	T7	T7
inducer	indoylacrylic acid	IPTG 60 µg/mL	IPTG	IPTG 0.67 mM	IPTG 1 mM	IPTG	IPTG 0.4 mM	IPTG 0.2 mM	IPTG 1 mM	IPTG (optimized protocol)
quantity	low amount of fusion protein /trpE protein	2: 12 mg TTFC/L (3–4% TPC)	11–14% TPC (with optimized promotor)	1 mg/L (0.5% TPC)	35 mg/L	un-specified	15–30% TPC (20–35 mg/mL after purification)	333 mg/L 42 L fermentor (46% TPC)	35% TCP	pET28a: 38 mg/mL pET22a: 32 mg/mL
solubility	soluble	1: low solubilty 2: soluble	soluble	soluble	soluble	soluble	soluble	soluble	soluble	soluble

## Data Availability

Data sharing not applicable.

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
