# Peer review of "Tetanus Toxin Fragment C: Structure, Drug Discovery Research and Production"

_pharmaceuticals, 2022, doi:10.3390/ph15060756_

Round 1

Reviewer 1 Report

Title: Tetanus Toxin Fragment C: Structure, Drug Discovery Research and Production

In this article, the authors have proposed a systematic review on Tetanus Toxin Fragment C (TTFC) by providing information about its structural features, properties, and its methods of production. Also, the authors have described the large uses of TTFC in drug discovery field. First, the authors have given concise introduction about tetanus toxin (TT) and Tetanus Toxin Fragment C (TTFC). Then, they have given structure and association with receptors. Next, the authors have described the TTFC properties and uses which includes TTFC neuronal properties (like permeability and CNS delivery, intrinsic neuronal protection, neurological properties) and immunological properties (like immunological properties against tetanus, enhancement of immunogenicity, in silico design of epitope-based vaccines). Also, the authors have given the list of in vivo uses of TTFC for neurological applications between 2005-2022. Finally, the authors have focused on TTFC production where, they explained papain digestion, TTFC production in recombinant system, Escherichia coli, TTFC expression and delivery in other bacterial host strains and TTFC expression in yeast and plant cells. Overall the review was well written and based on the importance of this topic, it can be published in Pharmaceuticals journal as it is.

Author Response

Comments on suggestions proposed by reviewer 1:

No suggestion proposed.

Response: The reviewer 1 did not propose any modification of the article. All authors thank reviewer 1 for the time spent reading our review article.

Reviewer 2 Report

Bayart and colleagues have provided an excellent, thorough review of the Hc fragment of tetanus toxin from a structural, biological, and production standpoint. The manuscript provides an excellent overview of its uses in neuroprotection and in vaccines.  The work is very well written and provides many references for further background reading. 

One slight suggestion: Please use the Society for Functional Glycomics representation of sugars to depict the sugars found in the gangliosides (Fig. 3). These can be found here: https://www.ncbi.nlm.nih.gov/glycans/snfg.html

Author Response

Suggestion from reviewer 2: Please use the Society for Functional Glycomics representation of sugars to depict the sugars found in the gangliosides (Fig. 3). These can be found here: https://www.ncbi.nlm.nih.gov/glycans/snfg.html

Response: We modified Figure 3 (page 4) as proposed reveiwer 2. All authors thank reviewer 2 for the time spent reading our review article.

Reviewer 3 Report

This paper deals with a very interesting issue which is the tetanus toxin fragment C (TTFC), including its structure, properties, production methods and its role in a wide variety of therapies. The review compiles the new findings about TTFC in the recent years, which makes it very relevant to the field as no review was published in the last years about TTFC.

The paper is well-written, clear and comprehensive. Tables and figures properly show the data and are easy to understand. However, I would like to provide some suggestions in order to improve the paper:

1. Table 2: in the column “Experimental model” different terminology is used to name the same experimental model. I would suggest to use the same terminology to make it easier to understand. For instance, “SOD1G93A mice” and “transgenic mice with the G93A human SOD1 mutation”.

2. Figure 5: bacteria scientific names in the legend should be in italics.

Author Response

Suggestion from reviewer 3:

1. Table 2: in the column “Experimental model” different terminology is used to name the same experimental model. I would suggest to use the same terminology to make it easier to understand. For instance, “SOD1G93A mice” and “transgenic mice with the G93A human SOD1 mutation”.

Response: Thanks for this comment and we did some modifications to harmonize corresponding terminologies (Pages 7 and 8).

2. Figure 5: bacteria scientific names in the legend should be in italics.

Response: Thanks for this comment and we did the optimization (Page 16).

All authors thank reviewer 3 for the time spent reading our review article.